# Analysis of 3-D Kinematics Using H-Gait System during Walking on a Lower Body Positive Pressure Treadmill

**DOI:** 10.3390/s21082619

**Published:** 2021-04-08

**Authors:** Yoshiaki Kataoka, Ryo Takeda, Shigeru Tadano, Tomoya Ishida, Yuki Saito, Satoshi Osuka, Mina Samukawa, Harukazu Tohyama

**Affiliations:** 1Faculty of Health Sciences, Hokkaido University, Sapporo 060-0812, Japan; aquarius_plus_150g@yahoo.co.jp (Y.K.); t.ishida@hs.hokudai.ac.jp (T.I.); h_u_d-of-h_s@eis.hokudai.ac.jp (Y.S.); osuka.satoshi@frontier.hokudai.ac.jp (S.O.); mina@eis.hokudai.ac.jp (M.S.); tohyama@med.hokudai.ac.jp (H.T.); 2Department of Rehabilitation, Health Sciences University of Hokkaido Hospital, Sapporo 002-8072, Japan; 3Faculty of Engineering, Hokkaido University, Sapporo 060-8628, Japan; 4National Institute of Technology, Hakodate College, Hakodate 042-8501, Japan; tadano@eng.hokudai.ac.jp

**Keywords:** wearable sensor, body weight support, positive pressure treadmill, gait, kinematics

## Abstract

Recently, treadmills equipped with a lower-body positive-pressure (LBPP) device have been developed to provide precise body weight support (BWS) during walking. Since lower limbs are covered in a waist-high chamber of an LBPP treadmill, a conventional motion analysis using an optical method is impossible to evaluate gait kinematics on LBPP. We have developed a wearable-sensor-based three-dimensional motion analysis system, H-Gait. The purpose of the present study was to investigate the effects of BWS by a LBPP treadmill on gait kinematics using an H-Gait system. Twenty-five healthy subjects walked at 2.5 km/h on a LBPP treadmill under the following three conditions: (1) 0%BWS, (2) 25%BWS and (3) 50%BWS conditions. Acceleration and angular velocity from seven wearable sensors were used to analyze lower limb kinematics during walking. BWS significantly decreased peak angles of hip adduction, knee adduction and ankle dorsiflexion. In particular, the peak knee adduction angle at the 50%BWS significantly decreased compared to at the 25%BWS (*p* = 0.012) or 0%BWS (*p* < 0.001). The present study showed that H-Gait system can detect the changes in gait kinematics in response to BWS by a LBPP treadmill and provided a useful clinical application of the H-Gait system to walking exercises.

## 1. Introduction

Recently, treadmills equipped with a lower-body positive-pressure (LBPP) device have been developed to provide precise body weight support (BWS) during walking [1]. These devices could correctly and easily reduce the force induced by ground reaction forces on the lower limbs during walking [2,3,4]. Additionally, users could walk on the LBPP treadmills more comfortably than on treadmills with a harness system [5]. This allowed for improvements in mobility functions during walking exercises. It was reported that BWS by the LBPP treadmill significantly reduced pain during walking and improved muscle strength in patients with knee osteoarthritis (OA) [6,7], and similarly with patients with hip OA [8].

The effects of BWS by the LBPP treadmill on lower limb kinematics are less well known. Since lower limbs are covered in a waist-high chamber of an LBPP treadmill, it is nearly impossible to evaluate gait kinematics on LBPP using conventional motion analysis with optical markers [9]. The effects of sex difference and interaction by BWS of LBPP on kinematics has not been clearly investigated except in reports of some differences in kinematics during walking [10,11,12]. Loverro et al. [13] detected sex-specific responses during walking under loaded conditions. They revealed that females and males change hip and knee mechanics differently while walking with a load. Therefore, sex difference and interaction by BWS of LBPP may appear on the gait kinematics of the lower limbs.

Recently, we have developed a wearable-sensor-based three-dimensional (3-D) motion analysis system, H-Gait [14]. This system can analyze the kinematics of lower limbs by seven wearable sensors that consist of tri-axial acceleration sensors and tri-axial gyro sensors without optical tracking. This H-Gait system may be able to calculate the 3-D kinematics of the lower limbs while walking on the LBPP treadmill. Therefore, the purpose of the present study was to investigate the effects of BWS by a LBPP treadmill on gait kinematics using the H-Gait system. In addition, the sex differences in gait kinematics under various BWS conditions by a LBPP treadmill during walking were examined as well.

## 2. Materials and Methods

### 2.1. Participants

Twenty-five healthy volunteers (13 males: age 25.2 ± 4.4 years old, height 170.8 ± 5.5 cm, body weight 64.1 ± 8.7 kg, 12 females: 25.1 ± 6.8 years old, height 159.3 ± 5.9 cm, body weight 54.7 ± 7.3 kg) were recruited in the present study. None of the participants had a history of bone fracture, surgery in the lower limbs or musculoskeletal-related disorders within the past 6 months or previous history of trauma.

### 2.2. H-Gait System

The wearable sensor system, the H-Gait system, calculates the 3-D orientation of the lower limbs using the body segment measurements of the lower limbs and the angular velocity and acceleration measurements during gait. Before calculating the 3-D lower limb orientations during gait, accurate lower limb segment measurements and initial sensor attachment errors must be established. In this work, spherical markers in combination with sensors were attached to the body during an inclined stance position to calibrate the sensor initial orientation to the body segment orientation.

Ten spherical foam polystyrene markers with a diameter of 2 cm were attached to the greater trochanters, the medial and lateral femoral condyle and the medial and lateral malleoli with double-sided tape. Three still images were taken from right, front and left sides of the participant in sequence by a 6-megapixel digital camera (EX-F1, CASIO COMPUTER CO., LTD, Tokyo, Japan). Subsequently, the length between the right and left greater trochanter, the thigh length (the greater trochanter to the lateral femoral condyle), the length of the lower leg (the lateral femoral condyle to the lateral malleoli) and the foot height (lateral malleoli to floor) were measured. Furthermore, we used the wire frame human gait model to quantify the lower limb posture during gait on the previous works by Tadano et al. [14]

Seven wearable sensor units (TSDN121, ATR-Promotions, Inc., Kyoto, Japan) that consisted of tri-axial acceleration sensors and tri-axial gyro sensors were placed on the pelvis, both thighs, both shanks and both feet of the participants (Figure 1). The participants wore custom tights with pockets, which stored the seven sensor units.

Acceleration and angular velocity data from seven sensors were collected simultaneously during walking via wireless Bluetooth in real time. Both acceleration and angular velocity were measured and synchronized at a sampling rate of 100 Hz. The acceleration and angular velocity were used to calculate the angular displacement. First, the acceleration data were used to measure the angular inclination angle of the sensors with respect to the gravitational acceleration. This was considered as the initial angle $\theta_{t}$. Afterwards, the subsequent angular displacement was calculated by integrating the angular velocity data ω with the sampling interval time (sampling rate of 100 Hz = sampling interval time of 0.01 s).
(1)$$\theta_{t}=\theta_{0}+\int_{0}^{t}\omega dt$$

These data were then processed in a MATLAB algorithm proposed by Takeda et al. [15] This algorithm could minimize accumulated signal noise, such as drift, through a combination of digital filters; sensor offset removal, and a robust double derivative and integration method. In addition, the sensor attachment errors were removed by conducting a calibration test before each gait trial. This calibration method involved, measuring the acceleration data of the sensors in the upright and inclined positions to calculate the initial inclination of each sensor with respect to the gravity. This allowed for the calculation of the initial 3-D orientation of the body segment to which the sensor unit was attached.

In relation to the validity and reliability of the gait analysis system, Tadano et al. [16] analyzed the kinematics of lower limbs in walking using an H-Gait system and compared with that of a camera-based motion analysis system. The correlation coefficient was 0.98 for the hip flexion angle, 0.97 for knee flexion angle and 0.78 for the ankle dorsiflexion angle, respectively. In addition, the minimal detectable changes on the spatiotemporal gait parameters and the peak hip/knee/ankle joint angles were as follows: 12.1 cm for the step length; 18.4 steps/min for the cadence; 5.1° for the peak hip flexion angle; 3.4°, peak hip extension angle; 3.9°, peak hip abduction angle; 2.5°, peak hip adduction angle; 7.0°, peak knee flexion angle; 3.5°, peak knee extension angle; 5.9°, peak ankle dorsiflexion angle; and 5.2°, peak ankle plantar flexion angle [8].

### 2.3. Walking Protocol

An LBPP treadmill (Anti-Gravity Treadmill M320, Alter G, Inc., Fremont, CA, USA) was used for the BWS (Figure 2). After participants wore specialized shorts for the LBPP treadmill, the height of the chamber was adjusted to the greater trochanter and fixed the dedicated shorts to the body. To determine the correlation of gravity and the internal pressure of the chamber, calibration was performed for each participant as previously reported [17]. Before walking on the LBPP treadmill, explanations were given and participants familiarized themselves to the LBPP treadmill for 3 min by walking. All participants walked at 2.5 km/h on a LBPP treadmill for 30 s that include the acceleration of walking under three conditions randomly. These BWS conditions were 0%BWS, 25%BWS and 50%BWS conditions. Participants were given 90 s to familiarize themselves to walking in each of the new BWS conditions.

### 2.4. Data Collection

Acceleration and angular velocity data from all participants were collected and post-processed. The subsequent 3-D orientations from the initial reference position were estimated by integrating the angular velocity with the drift removal method [15]. 3-D angular displacement from the initial upright position was calculated using a quaternion-based expression reported by Tadano et al. [16] For the gait cycle, one gait cycle from heel contact to the next heel contact was normalized to 100%. The swing and stance phases were defined using the heel contact and toe-off timings of both legs. The heel contact and toe-off timings were detected using the peak angular velocity data of the shank as previously reported [15,18].

The average value of 10 gait cycles during walking under each BWS condition was used as a representative value. Among the 3-D kinematic data under each BWS condition, the stride length, cadence, and peak flexion-extension angles of hip, knee and ankle joints and adduction–abduction angles of hip as well as knee joints on the dominant side during stance and swing phases were analyzed. The stride length and cadence were normalized to the participant’s leg length.

### 2.5. Statistical Analysis

Two-way ANOVAs with post hoc Bonferroni tests were performed to assess the effects of BWS and sex on spatiotemporal gait parameters and peak angles of hip, knee and ankle joints during swing and stance phases, respectively. The significant level was set at *p* < 0.05. Statistical analyses were performed using IBM SPSS Statistics version 17 (SPSS Inc., Chicago, IL, USA). In addition, the effect sizes for the main effect and the interaction between BWS and sex were calculated to indicate the magnitude of the differences using partial eta-squared (partial η^2^).

## 3. Results

### 3.1. The Effects of BWS on Spatiotemporal Gait Parameters

Regarding the stride length and the cadence, the main effects of BWS were observed (stride length: *p* < 0.001, partial η^2^ = 0.258, cadence: *p* < 0.001, partial η^2^ = 0.079). Stride length at the 50%BWS condition was significantly smaller than at the 0%BWS condition (*p* < 0.001) (Figure 3a). Cadence at the 50%BWS condition significantly increased compared with the 0%BWS condition (*p* = 0.038) as shown in Figure 3b.

### 3.2. The Effects of BWS on Kinematics during the Stance Phase

We also found that the condition of BWS changed kinematics in hip, knee and ankle joints for male and female subjects (Figure 4, Figure 5 and Figure 6). Concerning kinematics during the stance phase, the main effects of BWS on the peak hip extension and adduction angles were not observed (extension: *p* = 0.698, adduction: *p* = 0.444). The main effect of BWS on the peak knee adduction angle (*p* = 0.001, partial η^2^ = 0.172) was observed, although the effect of BWS on the peak knee extension angle (*p* = 0.072) was not observed (Figure 7c). The peak knee adduction angle at the 50%BWS condition significantly decreased compared to at the 25%BWS (*p* = 0.012) or 0%BWS condition (*p* < 0.001) (Figure 7d). Regarding the ankle joint, the main effect of BWS on the peak dorsiflexion angle was observed (*p* = 0.001, partial η^2^ = 0.172). The peak dorsiflexion angles at the 50%BWS (*p* = 0.002) and the 25%BWS conditions (*p* = 0.024) were significantly decreased compared to the 0%BWS condition (Figure 7e).

### 3.3. The Effects of BWS on Kinematics during the Swing Phase

Regarding kinematics during the swing phase, the main effects of BWS on the peak flexion and abduction angles of hip joint was observed (flexion: *p* = 0.004, partial η^2^ = 0.150, abduction: *p* = 0.001, partial η^2^ = 0.192). The peak hip flexion angle at the 50%BWS condition significantly decreased compared to the 0%BWS condition (*p* = 0.003) (Figure 8a). The peak hip abduction angle at the 50%BWS condition (*p* < 0.001) or the 25%BWS condition (*p* = 0.028) significantly decreased compared to the 0%BWS condition (Figure 8b). For the knee joint, the main effect of BWS on the peak knee flexion angle was observed (*p* < 0.001, partial η^2^ = 0.528) (Figure 8c), although the main effect of BWS on the peak knee abduction angle was not observed (*p* = 0.612) (Figure 8d). The peak knee flexion angle at the 50%BWS condition (*p* < 0.001) or the 25%BWS condition (*p* < 0.001) significantly decreased compared to the 0%BWS condition, and furthermore, the peak knee flexion angle at the 50%BWS condition significantly decreased compared to the 25%BWS condition (*p* = 0.004). There was a significant effect of BWS (*p* = 0.003, partial η^2^ = 0.156) on the peak ankle plantar flexion angle. The peak plantar flexion angles at the 50%BWS (*p* = 0.004) and the 25%BWS conditions (*p* = 0.047) significantly increased compared to the 0%BWS condition (Figure 8e).

### 3.4. The Effects of Sex on Spatiotemporal Gait Parameters and Kinematics

The main effect of sex on stride length (*p* < 0.001, partial η^2^ = 0.279) was observed, although the effect of sex on the cadence (*p* = 0.241) was not observed. There were no interactions between BWS and sex on spatiotemporal gait parameters.

Concerning kinematics during the stance phase, male participants were displaying significantly greater hip flexion than female participants (*p* = 0.015, partial η^2^ = 0.082) (Figure 7a), although the main effect of sex on the peak hip adduction angle was not observed (*p* = 0.242) (Figure 7b). Concerning the peak knee extension angle, the female participants were significantly larger than the male participants (*p* < 0.001, partial η^2^ = 0.191). Regarding the peak knee adduction angle, female participants were displayed significantly smaller results than male participants (*p* = 0.024, partial η^2^ = 0.071). For the peak dorsiflexion angle, female participants were displaying significantly larger results than male participants (*p* < 0.001, partial η^2^ = 0.211). There were no interactions between BWS and sex on kinematics during the stance phase.

Regarding kinematics during the swing phase, the effects of sex on the peak hip flexion and abduction angles were not detected (flexion: *p* = 0.162, abduction: *p* = 0.577). For the knee joint, the main effect of sex (*p* < 0.001, partial η^2^ = 0.198) on the peak flexion angle was observed, although the main effect of sex on the peak knee abduction angle was not observed (*p* = 0.346). There were significant effects of sex on the peak ankle plantar flexion angle (*p* < 0.001, partial η^2^ = 0.173). There were no interactions between BWS and sex on kinematics during the swing phase.

## 4. Discussion

BWS by a LBPP treadmill significantly affected the 3-D gait kinematics as well as the spatiotemporal gait parameters during walking using the analytical results obtained from the H-Gait system. BWS decreased peak adduction angles of hip and knee joints and the peak dorsiflexion angle of ankles during the stance phase. Additionally, BWS significantly decreased peak flexion angles in the hip and knee joints and significantly increased the peak plantar flexion angle in the ankle joint during the swing phase. There were some differences between females and males on gait kinematics. However, no interaction between BWS and sex on gait kinematics was found.

The present study revealed that BWS by a LBPP treadmill also affected the gait kinematics on the frontal plane during the stance phase. In particular, BWS decreased the peak adduction angle of knee joint. Patil et al. [4] showed that BWS by a LBPP treadmill tended to decrease the axial knee joint forces. It could be that BWS decreased the walking pain during the stance phase, this is because decreasing the peak knee adduction angle may result in a lesser load towards the medial knee compartment. Therefore, the effects of BWS on gait kinematics by a LBPP treadmill may involve improvements to relieving pain relating to knee osteoarthritis [6,7]. However, previous studies [6,7] that reported an improvement in pain relief during walking exercises using a LBPP treadmill do not take into account the influence of BWS on kinematics during walking. Therefore, it may be useful in the future to investigate the relationship between changes in kinematics and reduction in pain by BWS.

It has also been reported that BWS treadmill training also has been reported as a means to facilitate walking in subjects with neurological impairments [19,20]. In the present study, BWS decreases the peak angles during the swing phase. BWS allows for easier lower limb swinging, even at lesser peak flexion angles of hip and knee joints, which may facilitate the control of joint motion during walking. As a result, BWS training promotes motor learning in walking for patients with spinal cord injuries, which may be more effective for patients with spinal cord injuries [21].

There were some differences between females and males in the peak joint angles, and females especially showed more dorsiflexion of ankle joint and flexion of knee joints throughout the gait cycle compared to males in the present study. Since these sex differences were contrary to a previous report [22], the LBPP treadmill may have caused these differences. On the LBPP treadmill, the pelvis is immobilized in a chamber, which restricts pelvic movement. Because females have a greater pelvic rotation than males [22], the effect of pelvic restriction by the LBPP treadmill was greater in females than in males. The ankle and knee joints may have compensated for this restriction in females. Future studies should investigate the differences in kinematics, including sex differences, between walking on the LBPP treadmill and walking on the ground. On the other hand, there was no interaction between BWS and sex. Therefore, clinicians need not take into account sex differences when they apply BWS training to patients, because the effect of BWS was independent of sex. 

The H-Gait system uses gravity to determine the position and orientation of the segments, and considering that the LBPP treadmill uses a pressurized chamber to apply BWS and does not change gravity, it is believed that the H-Gait system accurately analyzes the gait kinematics on the LBPP treadmill. The H-Gait system can detect the changes in gait kinematics in response to BWS by a LBPP treadmill. Future studies should investigate the training effect of LBPP on gait kinematics among patients with injuries or after surgery.

There are several limitations to the present study. First, we targeted only healthy participants for the present study. These results may not be applicable to patients with injuries or after surgery. Additionally, we analyzed only the kinematics of lower limbs. Future studies are needed to examine the effects of BWS by LBPP treadmill on kinematics and kinetics of upper limbs and the trunk.

## 5. Conclusions

In conclusion, we investigated the effects of BWS by a LBPP treadmill on 3-D gait kinematics during walking in healthy participants using the H-Gait system. BWS by a LBPP treadmill significantly affected 3-D gait kinematics as well as spatiotemporal gait parameters during walking on an LBPP treadmill (Table 1). In particular, BWS by a LBPP treadmill decreased the peak adduction angles of knee joints (*p* = 0.001) and the peak dorsiflexion angle of ankle joints (*p* = 0.001) during the stance phase. Additionally, BWS significantly decreased peak flexion angles in the hip and knee joints (hip: *p* = 0.004, knee: *p* < 0.001) and significantly increased the peak plantar flexion angle in the ankle joint (*p* = 0.003) during the swing phase. The present study showed that the H-Gait system can detect the changes in gait kinematics in response to BWS by a LBPP treadmill and was proved to be a useful clinical application.

## Figures and Tables

**Figure 1 sensors-21-02619-f001:**
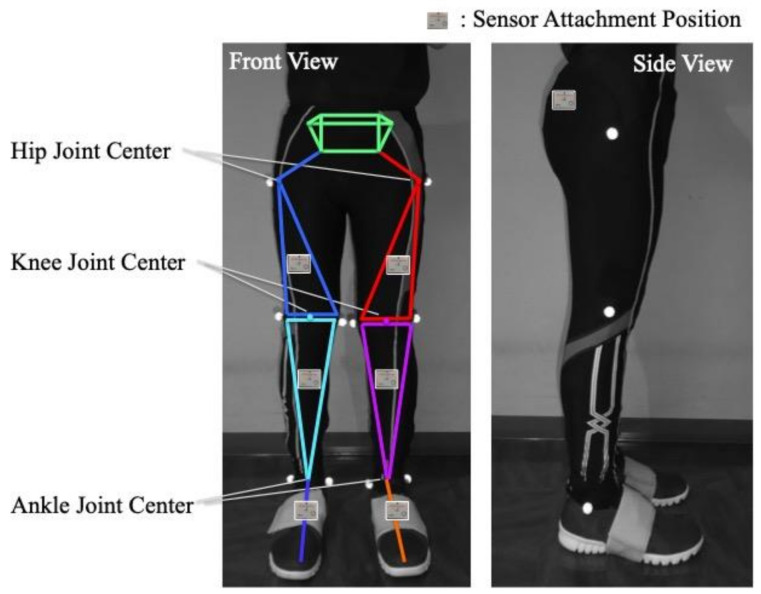
Sensor attachment locations. Sensor units are fixed to seven-body segments of the lower limbs and pelvis in the pockets of custom tights. Reflective markers were attached to anatomical landmarks of the lower limbs to calculate limb lengths.

**Figure 2 sensors-21-02619-f002:**
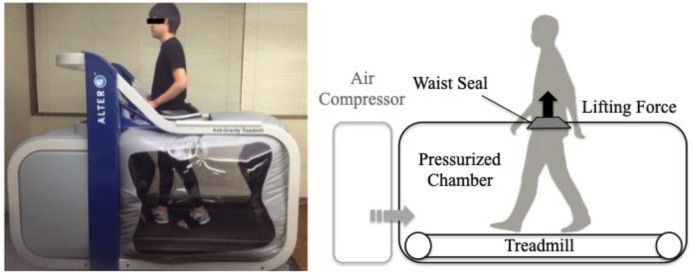
Lower body positive pressure (LBPP) treadmill used in the present study. In this treadmill, an air-compressor inflated the chamber to create a lifting force on the lower body by positive pressure. Subsequently, this treadmill could provide precise BWS during gait. Participants walked on this treadmill while wearing seven sensor units under the three BWS conditions.

**Figure 3 sensors-21-02619-f003:**
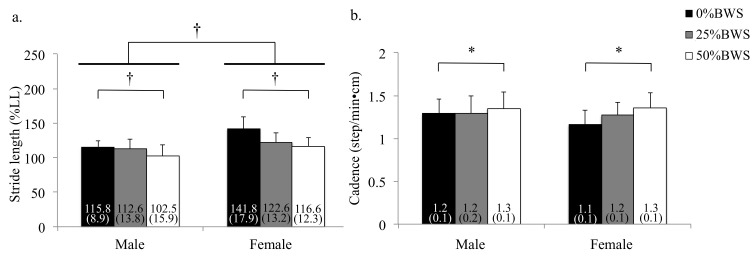
Spatiotemporal gait parameters under 0%, 25% and 50%BWS conditions for male and female participants ((**a**) stride length, (**b**) cadence). Data were reported as mean (standard deviation). * *p* < 0.05, † *p* < 0.01, BW: body weight, LL: leg length.

**Figure 4 sensors-21-02619-f004:**
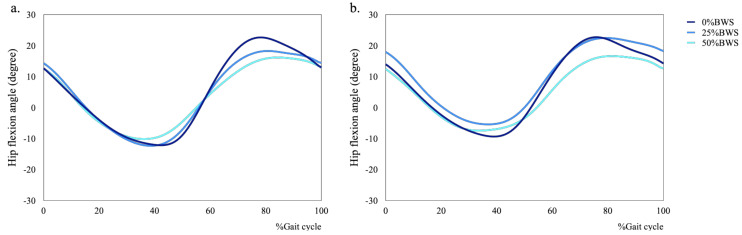
Mean plots of hip flexion during walking under 0%, 25% and 50%BWS conditions ((**a**) male, (**b**) female).

**Figure 5 sensors-21-02619-f005:**
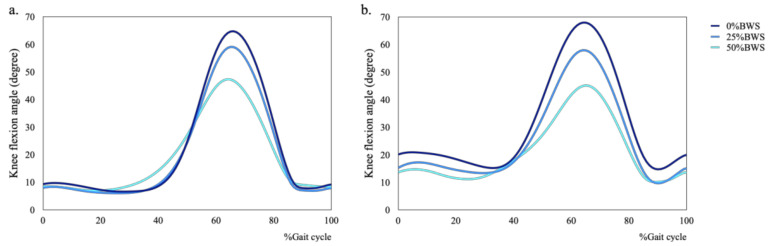
Mean plots of knee flexion during walking under 0%, 25% and 50%BWS conditions ((**a**) male, (**b**) female).

**Figure 6 sensors-21-02619-f006:**
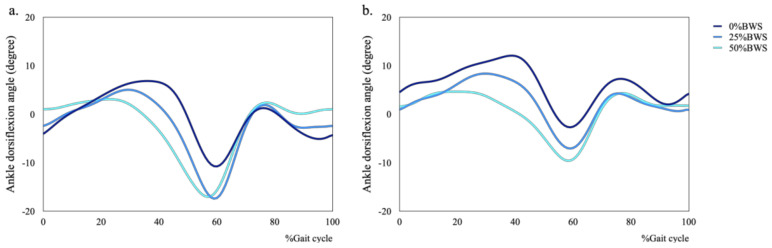
Mean plots of ankle dorsiflexion during walking under 0%, 25% and 50%BWS conditions ((**a**) male, (**b**) female).

**Figure 7 sensors-21-02619-f007:**
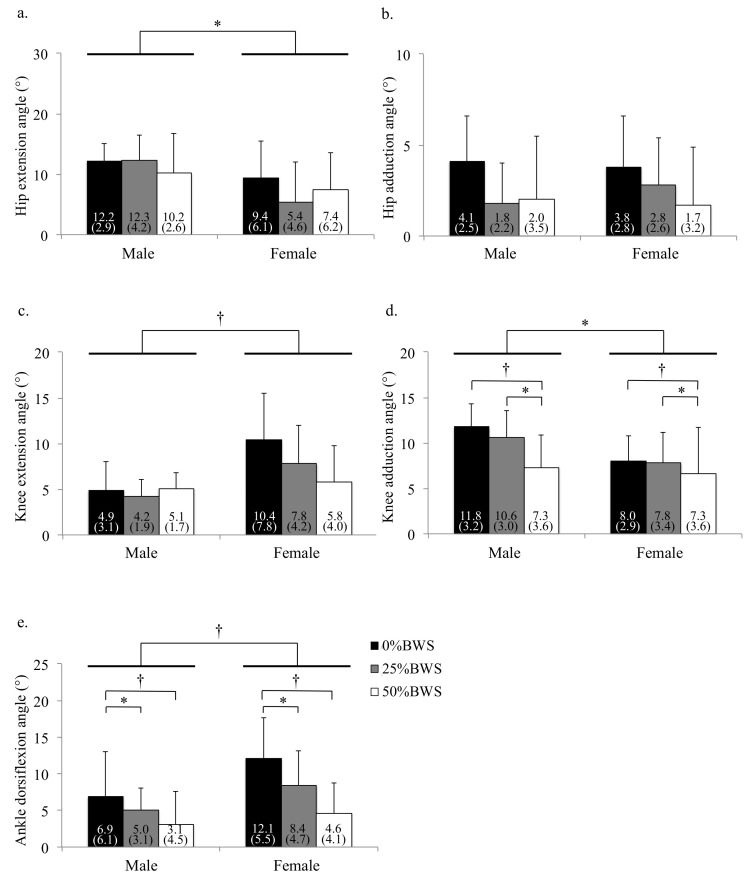
The peak angle of each joint during stance phase under 0%, 25% and 50%BWS conditions for male and female participants ((**a**) hip extension, (**b**) hip adduction, (**c**) knee extension, (**d**) knee adduction, (**e**) ankle dorsiflexion). Data are reported as mean (standard deviation). * *p* < 0.05, † *p* < 0.01, BW: body weight.

**Figure 8 sensors-21-02619-f008:**
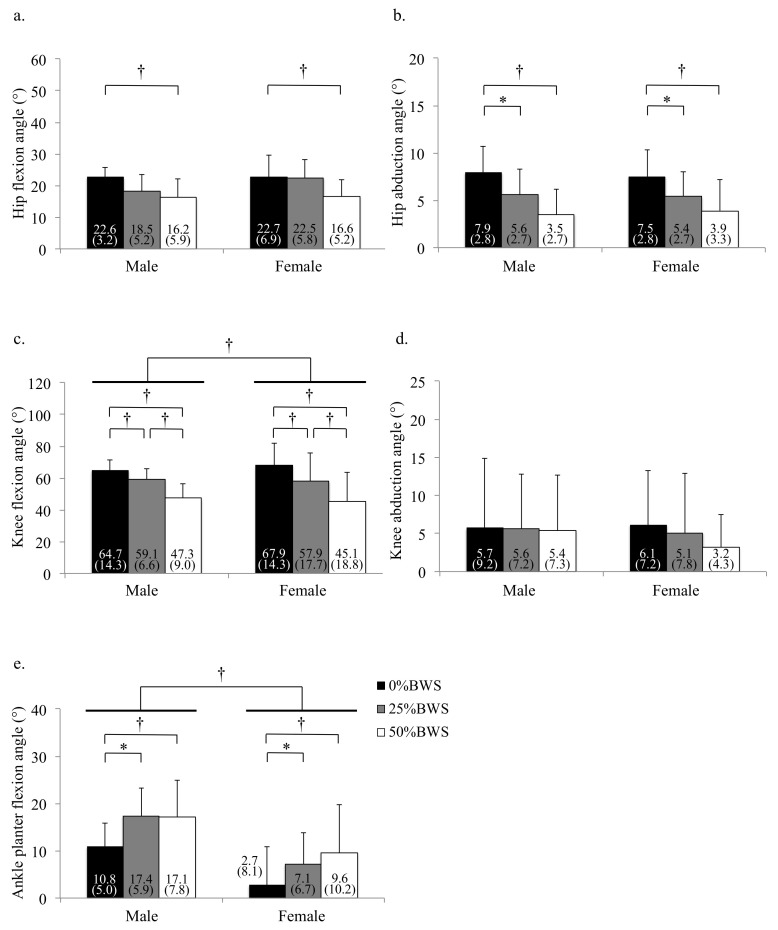
The peak angle of each joint during swing phase under 0%, 25% and 50%BWS conditions for male and female participants ((**a**) hip flexion, (**b**) hip abduction, (**c**) knee flexion, (**d**) knee abduction, (**e**) ankle plantar flexion). Data are reported as mean (standard deviation). * *p* < 0.05, † *p* < 0.01, BW: body weight.

**Table 1 sensors-21-02619-t001:** Peak angles of the hip, knee, and ankle joints under 0%, 25% and 50%BWS conditions for male and female participants.

Variables		0%BWS	25%BWS	50%BWS	Main Effect: BWS (*p*-Value)
Knee adduction (°)	Male Female	11.8 (3.2) 8.0 (2.9)	10.6 (3.0) 7.8 (3.4)	7.3 (3.6) *^,†^ 7.3 (3.6) *^,†^	0.001
Ankle dorsiflexion (°)	Male Female	6.9 (6.1) 12.2 (5.5)	5.0 (3.1) ^#^ 8.4 (4.7) ^#^	3.1 (4.5) ^†^ 4.6 (4.1) ^†^	0.001
Hip flexion (°)	Male Female	22.6 (3.2) 22.7 (6.9)	18.5 (5.2) 22.5 (5.8)	16.2 (5.9) ^†^ 16.6 (5.2) ^†^	0.004
Knee flexion (°)	Male Female	64.7 (14.3) 67.9 (14.3)	59.1 (6.6) ^†^ 57.9 (17.7) ^†^	47.3 (9.0) ^†,^^$^ 45.1 (18.8) ^†,^^$^	<0.001
Ankle plantar flexion (°)	Male Female	10.8 (5.0) 2.7 (8.1)	17.4 (5.9) ^#^ 7.1 (6.7) ^#^	17.1 (7.8) ^†^ 9.6 (10.2) ^†^	0.003

Data are presented as mean (standard deviation). BWS: body weight support. *: <0.05 (vs. 25%BWS), ^$^: <0.01 (vs. 25%BWS), ^†^: <0.01 (vs. 0%BWS), ^#^: <0.05 (vs. 0%BWS).

## Data Availability

The datasets used and/or analyzed during the current study are available from the corresponding author on reasonable request.

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
