# Peer review of "Analysis of 3-D Kinematics Using H-Gait System during Walking on a Lower Body Positive Pressure Treadmill"

_sensors, 2021, doi:10.3390/s21082619_

Round 1

Reviewer 1 Report

The authors investigated the effects of BWS by a LBPP treadmill on 53 gait kinematics using H-Gait system. Their results are supported by their experiment results. This paper is suitable for a communication paper. The reviewer just has few questions:

1) There is difference between the effects of BWS on females and males, especially in Figure 6, could the author give a discussion about that?

2)could the author explain a little bit about how they collected and calculate the angular displacement in their experiemnt? there is just a reference about that, which is not so clear for readers to directly understand their methods.

Author Response

We wish to express our appreciation to the reviewers for their insightful comments on our manuscript. We have responded to each of the reviewers’ comments and suggestions. Additionally, we have revised the English in the manuscript thoroughly checked and edited for language and form.

Reviewer reports:

The authors investigated the effects of BWS by a LBPP treadmill on gait kinematics using H-Gait system. Their results are supported by their experiment results. This paper is suitable for a communication paper. The reviewer just has few questions:

Comments:

  1. There is difference between the effects of BWS on females and males, especially in Figure 6, could the author give a discussion about that?

Authors’ response and reaction: Thank you for your valuable suggestion. The present study found some sex main effects on the peak joint angles during gait. For stance phase, female participants showed significantly larger peak knee extension and ankle dorsiflexion angles and smaller peak hip flexion and knee adduction angles than male participants. For swing phase, female participants showed significantly larger peak knee flexion and smaller peak plantar flexion than male participants. However, there was no interaction between BWS and sex on the peak joint angles in the present study. Therefore, the effects of BWS by LBPP on the peak joint angles during gait would not be differed between male and female participants in the present study. We have revised the Discussion section following the reviewer’s suggestion.

Page 13, lines 348–360

There were some differences between females and males in the peak joint angles, especially females showed more dorsiflexion of ankle joint and flexion of knee joint throughout the gait cycle compared to males in the present study. Since these sex differences were contrary to a previous report [22], the LBPP treadmill may have caused these differences. On the LBPP treadmill, the pelvis is immobilized in a chamber, which restricts pelvic movement. Because females have greater pelvic rotation than males [22], the effect of pelvic restriction by the LBPP treadmill was greater in females than in males. The ankle and knee joints may have compensated for this restriction in female. Future studies should investigate the differences in kinematics, including sex differences, between walking on the LBPP treadmill and walking on the ground. On the other hand, there was no interaction between BWS and sex. Therefore, clinicians need not take into account of sex difference when they apply the BWS training for patients, because the effect of BWS was independent of sex.

  1. Could the author explain a little bit about how they collected and calculate the angular displacement in their experiment? There is just a reference about that, which is not so clear for readers to directly understand their methods.

Authors’ response and reaction: Thank you for the question concerning the method for calculating the angular displacement. The response to this has been summarized as below. There are more technical details into how various matrices were used to convert the various sensor, body segment and global coordinate systems and the drift removal procedures. However, due to the word count limitation, we believe that there is not enough space in the manuscript to include this. We hope that explanation above is sufficient, if not the authors will be more than glad to go into the more technical details.

Page 2-3, lines 91–100

The acceleration and angular velocity were used to calculate the angular displacement. First, the acceleration data was used to measure the angular inclination angle of the sensors with respect to the gravitational acceleration. This was considered as the initial angle . Afterwards, the subsequent angular displacement was calculated by integrating the angular velocity data  with the sampling interval time (sampling rate of 100Hz = sampling interval time of 0.01s).

Reviewer 2 Report

This study objectively examines the effects of body weight support (BWS) on lower limb kinematics during walking. Twenty-five healthy subjects (HS) walked at 2.5 km/h on a lower-body positive pressure (LBPP) treadmill under 3 different BWS conditions (i.e. 0, 25, 50% BWS) while monitored through a wearable sensor system (i.e. H-Gait system). The H-Gait system consisted of 7 inertial sensors composed of three-axial accelerometers and gyroscopes. Several kinematic measures, including the stride length, cadence, peak flexion-extension angles of hip, knee and ankle joints, adduction-abduction angles of hip and knee joints, were assessed both during stance and swing phases. The authors identified a number of kinematic changes of walking under different BWS conditions as well as with respect to subjects’ sex. The authors concluded that the H-Gait system may help to detect changes in gait kinematics in response to BWS.

The topic of the study is of interest. Also, the manuscript is well written and organized. I have a few minor comments for the authors:

- The authors should improve and expand the discussion of findings, including those concerning the effect of subjects’ sex.

- In order to improve the readability of the text, I would suggest presenting results on BWS and sex effects separately in two different paragraphs.

- Examining possible clinical prospects of BWS also in specific neurological disorders (e.g., patients with spinal cord injuries) would add value to the manuscript.   

Author Response

We wish to express our appreciation to the reviewers for their insightful comments on our manuscript. We have responded to each of the reviewers’ comments and suggestions. Additionally, we have revised the English in the manuscript thoroughly checked and edited for language and form.

Reviewer reports:

This study objectively examines the effects of body weight support (BWS) on lower limb kinematics during walking. Twenty-five healthy subjects (HS) walked at 2.5 km/h on a lower-body positive pressure (LBPP) treadmill under 3 different BWS conditions (i.e. 0, 25, 50% BWS) while monitored through a wearable sensor system (i.e. H-Gait system). The H-Gait system consisted of 7 inertial sensors composed of three-axial accelerometers and gyroscopes. Several kinematic measures, including the stride length, cadence, peak flexion-extension angles of hip, knee and ankle joints, adduction-abduction angles of hip and knee joints, were assessed both during stance and swing phases. The authors identified a number of kinematic changes of walking under different BWS conditions as well as with respect to subjects’ sex. The authors concluded that the H-Gait system may help to detect changes in gait kinematics in response to BWS.

The topic of the study is of interest. Also, the manuscript is well written and organized. I have a few minor comments for the authors:

Comments:

  1. The authors should improve and expand the discussion of findings, including those concerning the effect of subjects’ sex.

Authors’ response and reaction: Thank you for your valuable suggestion. We improve and expand the discussion about the effect of sex and interaction between BWS and sex. We have revised the Discussion section following the reviewer’s suggestion.

Page 13, lines 341–347

It has also been reported that BWS treadmill training also has been reported as a means to facilitate walking in subjects with neurological impairments [19, 20]. In the present study, BWS decreases the peak angles during the swing phase. BWS allows for easier lower limb swinging even at less peak flexion angles of hip and knee joints, which may facilitate the control of joint motion during walking. As a result, BWS training promotes motor learning in walking for patients with spinal cord injuries, which may be more effective for patients with spinal cord injuries [21].

Page 13, lines 348–360

There were some differences between females and males in the peak joint angles, especially females showed more dorsiflexion of ankle joint and flexion of knee joint throughout the gait cycle compared to males in the present study. Since these sex differences were contrary to a previous report [22], the LBPP treadmill may have caused these differences. On the LBPP treadmill, the pelvis is immobilized in a chamber, which restricts pelvic movement. Because females have greater pelvic rotation than males [22], the effect of pelvic restriction by the LBPP treadmill was greater in females than in males. The ankle and knee joints may have compensated for this restriction in female. Future studies should investigate the differences in kinematics, including sex differences, between walking on the LBPP treadmill and walking on the ground. On the other hand, there was no interaction between BWS and sex. Therefore, clinicians need not take into account of sex difference when they apply the BWS training for patients, because the effect of BWS was independent of sex.

  1. In order to improve the readability of the text, I would suggest presenting results on BWS and sex effects separately in two different paragraphs.

Authors’ response and reaction: Thank you for your careful reading. We have revised the Results section in different paragraphs following the reviewer’s suggestion.

Page 4, lines 169

3.1. The effects of BWS on spatiotemporal gait parameters

Page 5, lines 187

3.2. The effects of BWS on kinematics during the stance phase

Page 7, lines 220

3.3. The effects of BWS on kinematics during the swing phase

Page 8, lines 244

3.4. The effects of sex on spatiotemporal gait parameters and kinematics

  1. Examining possible clinical prospects of BWS also in specific neurological disorders (e.g., patients with spinal cord injuries) would add value to the manuscript.   

Authors’ response and reaction: Thank you for your valuable suggestion. We have revised the Discussion section about clinical prospects of BWS for patients with spinal cord injuries following the reviewer’s suggestion.

Page 13, lines 341–347

It has also been reported that BWS treadmill training also has been reported as a means to facilitate walking in subjects with neurological impairments [19, 20]. In the present study, BWS decreases the peak angles during the swing phase. BWS allows for easier lower limb swinging even at less peak flexion angles of hip and knee joints, which may facilitate the control of joint motion during walking. As a result, BWS training promotes motor learning in walking for patients with spinal cord injuries, which may be more effective for patients with spinal cord injuries [21].

Reviewer 3 Report

Good day,

Thank you for an interesting article.

The topic is well presented and explained.

I found a typo in line 166. I think it should say "Male participants were displaying significantly..."

To increase the readability of this fine article even further I suggest the following:

  • Both paragraphs from line 165-199 are hard to follow. It is easy to get lost in numbers and terms here. Please include angle illustrations which are mentioned here, and separate the paragraphs with it
  • In conclusions it would be good to use some result numbers instead of just saying 'significantly' and to include a short summary results table.

Kind reagrds,

David

Author Response

We wish to express our appreciation to the reviewers for their insightful comments on our manuscript. We have responded to each of the reviewers’ comments and suggestions. Additionally, we have revised the English in the manuscript thoroughly checked and edited for language and form.

Reviewer reports:

Thank you for an interesting article.

The topic is well presented and explained.

Comments:

  1. I found a typo in line 166. I think it should say "Male participants were displaying significantly...":

Authors’ response and reaction: Thank you for your careful reading. We have revised the Results section following the reviewer’s suggestion.

Page 8-9, lines 249–255

Concerning kinematics during the stance phase, male participants were displaying significantly greater hip flexion than female participants (P = 0.015, partial η2 = 0.082) (Figure 7a), although the main effect of sex on the peak hip adduction angle was not observed (P = 0.242) (Figure 7b).

Page 9, lines 257–260

Regarding the peak knee adduction angle, female participants were displaying significantly smaller than male participants (P = 0.024, partial η2 = 0.071).

Page 9, lines 263–265

For the peak dorsiflexion angle, female participants were displaying significantly bigger than male participants (P < 0.001, partial η2 = 0.211)

  1. To increase the readability of this fine article even further I suggest the following:

Both paragraphs from line 165-199 are hard to follow. It is easy to get lost in numbers and terms here. Please include angle illustrations which are mentioned here, and separate the paragraphs with it

Authors’ response and reaction: Thank you for your careful reading and valuable suggestion. We have revised the Results section following the reviewer’s suggestion to increase the readability of our study.

Page 4, lines 169

3.1. The effects of BWS on spatiotemporal gait parameters

Page 5, lines 187

3.2. The effects of BWS on kinematics during the stance phase

Page 7, lines 220

3.3. The effects of BWS on kinematics during the swing phase

Page 8, lines 244

3.4. The effects of sex on spatiotemporal gait parameters and kinematics

  1. In conclusions it would be good to use some result numbers instead of just saying 'significantly' and to include a short summary results table.

Authors’ response and reaction: Thank you for your valuable suggestion. We added P-value in the conclusions and a short summary results table in the conclusion following the reviewer’s suggestion.

Page 13, lines 375–384

BWS by a LBPP treadmill significantly affected 3-D gait kinematics as well as spatiotemporal gait parameters during walking on a LBPP treadmill (Table 1). In particular, BWS by a LBPP treadmill decreased peak adduction angles of knee joint (P = 0.001) and peak dorsiflexion angle of ankle joint (P = 0.001) during the stance phase. Additionally, BWS significantly decreased peak flexion angles in the hip and knee joints (hip: P = 0.004, knee: P < 0.001) and significantly increased the peak plantar flexion angle in the ankle joint (P = 0.003) during the swing phase.

Page 14, lines 386–389

Table 1. Peak angles of the hip, knee, and ankle joints under 0%, 25% and 50%BWS conditions for male and female participants

Variables

0%BWS

25%BWS

50%BWS

Main effect: BWS (P-value)

Knee adduction (°)

Male

Female

11.8 (3.2)

8.0 (2.9)

10.6 (3.0)

7.8 (3.4)

7.3 (3.6)*†

7.3 (3.6)*†

0.001

Ankle dorsiflexion (°)

Male

Female

6.9 (6.1)

12.2 (5.5)

5.0 (3.1)#

8.4 (4.7)#

3.1 (4.5)

4.6 (4.1)

0.001

Hip flexion (°)

Male

Female

22.6 (3.2)

22.7 (6.9)

18.5 (5.2)

22.5 (5.8)

16.2 (5.9)

16.6 (5.2)

0.004

Knee flexion (°)

Male

Female

64.7 (14.3)

67.9 (14.3)

59.1 (6.6)

57.9 (17.7)

47.3 (9.0)$

45.1 (18.8)$

<0.001

Ankle plantar flexion (°)

Male

Female

10.8 (5.0)

2.7 (8.1)

17.4 (5.9)#

7.1 (6.7)#

17.1 (7.8)

9.6 (10.2)

0.003

Data are presented as mean (standard deviation). BWS: body weight support

*: <0.05 (vs 25%BWS), $: <0.01 (vs 25%BWS), : <0.01 (vs 0%BWS), #: <0.05 (vs 0%BWS)